

# Size-resolved isotope analysis reveals anthropogenic reactive nitrogen transport and transformation in Taiwan mountain forests

Wen-Chien Lee,[1] Ming-Hao Huang,[1] Wei-Chieh Huang,[1] Jen-Ping Chen,[1] Yen-Jen Lai,[2,3] Haojia Ren,[4]* and Hui-Ming Hung[1]*

[1] Department of Atmospheric Sciences, National Taiwan University, Taipei, 10617, Taiwan

[2] Experimental Forest, National Taiwan University, Nantou, 557004, Taiwan

[3] Department of Agricultural Chemistry, National Taiwan University, Taipei, 10617, Taiwan

[4] Department of Geosciences, National Taiwan University, Taipei, 10617, Taiwan

*Correspondence to*: Haojia Ren (abbyren@ntu.edu.tw) and Hui-Ming Hung (hmhung@ntu.edu.tw)

**Abstract.**

Reactive nitrogen (Nr) species such as particulate ammonium ($p$NH$_4^+$) and nitrate ($p$NO$_3^-$) cause air pollution and affect ecosystems, yet their transformation processes in mountain forests are not well-characterized. Size-resolved isotope analysis of aerosols could reveal these processes, but is rarely performed due to low particle concentrations. We overcame this limitation by combining size-segregated aerosol sampling at Xitou, Taiwan, with sensitive isotopic techniques and Bayesian modeling. Functional groups were analyzed by Fourier-transform infrared spectroscopy (FTIR-ATR), and isotopes $\delta^{15}$N and $\delta^{18}$O were measured by gas chromatography-isotope ratio mass spectrometry (GC-IRMS), enabling quantification of $p$NH$_4^+$ source contributions and $p$NO$_3^-$ formation pathways. Typical diurnal patterns, with higher daytime particle concentrations, were disrupted during a 26-hour fog caused by stagnant atmospheric conditions. During fog, the average $\delta^{15}$N-NH$_4^+$ decreased from 11.75±2.42‰ (mean±1σ) during clear periods to 7.75±1.37‰, while $\delta^{15}$N-NO$_3^-$ dropped from −2.57±1.80‰ to −4.51±1.79‰, indicating continued isotopic fractionation under reduced urban influence. Size-resolved isotope results revealed nitrate evolution during transport: urban plumes retained O$_3$-driven oxidation signatures with isotopic fractionation, whereas mountain-formed nitrate was produced via RO$_2$-involved processes with greater isotopic fractionation and enhanced biogenic contributions. Bayesian modeling indicated that 50−83% of NH$_3$ emissions originated from combustion-related sources, while 42−95% of $p$NO$_3^-$ formed through RO$_2$-initiated oxidation during daytime and 6−84% through heterogeneous reactions at night. These findings emphasize the importance of controlling urban NO$_x$ and combustion-related NH$_3$ emissions to reduce downwind Nr pollution and demonstrate how size-resolved isotope analysis elucidates aerosol evolution along transport pathways.

## 1. Introduction

Anthropogenic activities have increased reactive nitrogen (Nr) in the Earth system, contributing to climate change, biodiversity loss, acid deposition, and air pollution. Among Nr species, particulate ammonium ($p$NH$_4^+$) and nitrate ($p$NO$_3^-$) derived from ammonia (NH$_3$) and nitrogen oxides (NO$_x$) are major pollutants that degrade air quality, reduce visibility, and increase human morbidity (Gong et al., 2024; Zhang et al., 2017). In Asian countries, these species account for approximately 10−37% of non-refractory PM$_1$ (particulate matter with a diameter less than 1 μm) mass (Zhou et al., 2020). Although anthropogenic Nr is transported to rural areas, its distribution remains uneven, exacerbating regional disparities in nitrogen availability and environmental impacts (Galloway et al., 2008).



Therefore, understanding the formation, transport, and sources of Nr is essential for evaluating its origins and environmental impacts.

Nr is emitted from both anthropogenic and biogenic sources. $NH_3$ is predominantly released from agricultural activities and is primarily removed through dry deposition, precipitation scavenging, and chemical conversion to $pNH_4^+$ via reactions with acidic products from $NO_x$ and sulfur dioxide ($SO_2$) oxidations (Meng et al.,

2017). Sources of $NO_x$ include coal combustion, vehicle exhausts, biomass burning, and soil emission (Fan et al., 2020). $NO_x$ is removed from the atmosphere through oxidation processes (Romer et al., 2016). In the presence of sunlight, atmospheric $NO_x$ rapidly transforms through ozone ($O_3$) or organic peroxy radical ($RO_2$) pathways before converting to nitric acid ($HNO_3$) as follows:

$$NO + O_3 \left(RO_2\right) \rightarrow NO_2 + O_2 \left(RO\right) \tag{R1}$$

$$NO_2 + hv \rightarrow NO + O \tag{R2}$$

$$NO_2 + OH + M \rightarrow HNO_3 + M \tag{R3}$$

Tropospheric $O_3$ is a key oxidant formed when volatile organic compounds (VOCs) undergo oxidation in the presence of $NO_x$ and sunlight (Haagen-Smit et al., 1953). Another important oxidant, $RO_2$ radicals, primarily form during daytime when hydroxyl radicals (OH) react with biogenic VOCs such as isoprene in forested environments

(Romer et al., 2016). In the absence of sunlight, $NO_x$ accumulates as nitrogen dioxide ($NO_2$) in the presence of $O_3$, which then undergoes heterogeneous reactions to produce $HNO_3$ as follows:

$$NO_2 + O_3 \rightarrow NO_3 + O_2 \tag{R4}$$

$$NO_3 + NO_2 + M \rightarrow N_2O_5 + M \tag{R5}$$

$$N_2O_5 + H_2O \rightarrow 2HNO_3 \tag{R6}$$

These oxidation processes not only regulate atmospheric chemistry but also leave distinct isotopic signatures, making stable isotope analysis a powerful tool for tracing the origin, evolution, and transport of aerosol particles (Moore, 1977). The sources of $pNH_4^+$ can be distinguished using nitrogen isotope ratios ($\delta^{15}N$), as different emission sources exhibit characteristic $\delta^{15}N$ signatures (Savard et al., 2017). Additionally, the oxidation pathways leading to $pNO_3^-$ formation can be inferred from oxygen isotope ratios ($\delta^{18}O$), since each oxidant ($O_3$, OH, $RO_2$) imparts a

unique $\delta^{18}O$ signature (Walters and Michalski, 2016).

The sources of $pNH_4^+$ and the formation mechanisms of $pNO_3^-$ have successfully been identified using isotopic Bayesian mixing model frameworks (e.g., IsoSource, SIAR, or MixSIAR) (Fan et al., 2020; Chang et al., 2018; Kawashima et al., 2023; Pan et al., 2016). However, these processes are spatially variable, and relatively little is known about the sources and atmospheric processing of Nr in East Asian mountain forests, where complex

interactions between local emissions and long-range transported pollutants are common (Guha et al., 2017). During atmospheric transport, nitrogen isotope ratios undergo modifications due to deposition, chemical transformation, and photochemical processes. In particular, the gas-particle partitioning between $NH_3$ and $pNH_4^+$ can lead to a progressive decrease in $\delta^{15}N-NH_4^+$ values with increasing transport distance (Pan et al., 2016). Similarly, $pNO_3^-$ experiences gradual $\delta^{15}N$ depletion during long-range transport as kinetic fractionation during $HNO_3$ uptake

preferentially incorporates lighter isotopes into the particulate phase (Gobel et al., 2013; Walters and Michalski, 2016). However, due to the typically low concentrations of aerosol particles, detailed $\delta^{15}N$ and $\delta^{18}O$ analyses of size-segregated aerosol particles remain limited, constraining our understanding of emission sources and size-dependent nitrogen transformation pathways (Morin et al., 2009). Previous work at Xitou has characterized Nr sources using $\delta^{15}N$ and $\delta^{18}O$ isotope signatures (Chen, T. Y. et al., 2022), yet the role of prolonged fog due to suppressed





atmospheric transport was not well investigated. The prolonged fog can significantly alter aerosol composition and
      isotopic signals by promoting aqueous-phase reactions and enhancing local gas-particle partitioning.

      In this study, we investigate the size-dependent isotope distribution of aerosols and examine a rare ~26-hour
      fog episode to understand how stagnant and high-RH conditions modulate the transformation and distribution
      patterns of $p\mathrm{NH_4^+}$ and $p\mathrm{NO_3^-}$. Using stable isotope analysis with Bayesian source apportionment modeling, we
investigate the relative contributions of local and transported sources and evaluate how fog conditions affect the
      isotopic composition of Nr species in a subtropical mountain forest. This study provides valuable insights into Nr
      sources and atmospheric transformation processes, which are essential for informing effective air quality
      management strategies.

## 2. Methods

A field measurement was conducted from 17 to 24 April 2021, at the Xitou Experimental Forest of National
      Taiwan University (23°40'12" N, 120°47'54" E, 1179 m above sea level; Fig. S1). The measurement site is located
      in a river valley adjacent to the western foothills of Taiwan's Central Mountain Range, approximately 65−75 km
      southeast of the Taichung metropolitan area and 50−60 km inland from the Taiwan Strait. The area is characterized
      by limited direct anthropogenic influence, with no significant industrial facilities, oil refineries, or large-scale
farming operations, though some small-scale agriculture activities exist locally. During the measurement period,
      size-segregated aerosol particles were collected for daytime and nighttime intervals to analyze their chemical
      composition and the isotopic signatures ($\delta^{15}\mathrm{N}$ and $\delta^{18}\mathrm{O}$) of Nr species. Meanwhile, meteorological parameters and
      atmospheric conditions were monitored. The isotopic data $\delta^{15}\mathrm{N}\text{-}\mathrm{NH_4^+}$ and $\delta^{18}\mathrm{O}\text{-}\mathrm{NO_3^-}$ were further applied to
      estimate the sources and formation pathways of Nr aerosols using a Mixed Stable Isotope Analysis in R (MixSIAR)
framework (Stock et al., 2018).

### 2.1. Sample collection

      Size-segregated aerosol samples were collected using a micro-orifice uniform deposit impactor (MOUDI,
      Model 125R, MSP Corporation, Shoreview, Minnesota, USA) equipped with 46.2 mm polytetrafluoroethylene
      (PTFE) filters (Whatman 7592-104). The MOUDI was operated at a flow rate of 30 L min⁻¹, providing aerodynamic
cut-point diameters of 0.056, 0.1, 0.18, 0.32, 0.56, 1.0, 1.8, 3.2, 5.6, and 10 μm. Sampling was conducted on a half-
      day basis, with daytime (09:00 to 17:00 LT, denoted as 'D') and nighttime (18:00 to 06:00 LT on the following day,
      denoted as 'N') periods. Each sample was labeled as 'ddD' or 'ddN', where 'dd' represents the day of the month.
      After sample collection, filters were sealed in aluminum foil and stored at 4°C until analysis. During the sampling
      period, a custom-built Air Quality Box (AQB) was applied to continuously monitor meteorological parameters,
including temperature, relative humidity (RH), and trace gas concentrations (CO, NO, NO₂, and $O_x$ (NO₂ + O₃)).
      However, due to the stability of the calibration issue, only CO is considered in this study (Huang, W. C. et al., 2024).
      Additionally, visibility, radiation, wind speed, and wind direction data were acquired from the Agricultural
      Meteorological Station of the Experimental Forest, College of Bio-Resources and Agriculture, National Taiwan
      University.

### 2.2. Sample analysis

      Sample analysis consisted of three sequential steps: concentration measurement, extraction, and isotopic
      measurements (Fig. 1). First, attenuated total reflectance Fourier transform infrared spectroscopy (ATR-FTIR,
      Nicolet 6700, Thermo Fisher Scientific, Madison, WI, USA) was used to quantify sample concentrations on the
      filters to ensure sufficient nitrogen content ($\geq 1$ μM N as $\mathrm{NO_3^-} + \mathrm{NH_4^+}$) for size-resolved isotope analysis, with
details in Supplementary Description S1. Filters from adjacent size bins collected during the same period were
      combined when samples contained insufficient N content. The combined filters, with mass-weighted particle
      diameter calculated following Supplementary Description S2, were extracted in 30 mL Milli-Q water (18.2 MΩ at
      25°C) using 30-min ultrasonication, and extracts were filtered through 0.22 μm Millipore syringe filters and stored
      in high-density polyethylene (HDPE) bottles. The extracted nitrogen species were oxidized to nitrate ion ($\mathrm{NO_3^-}$)
using potassium persulfate reagent for total nitrogen (TN) analysis.



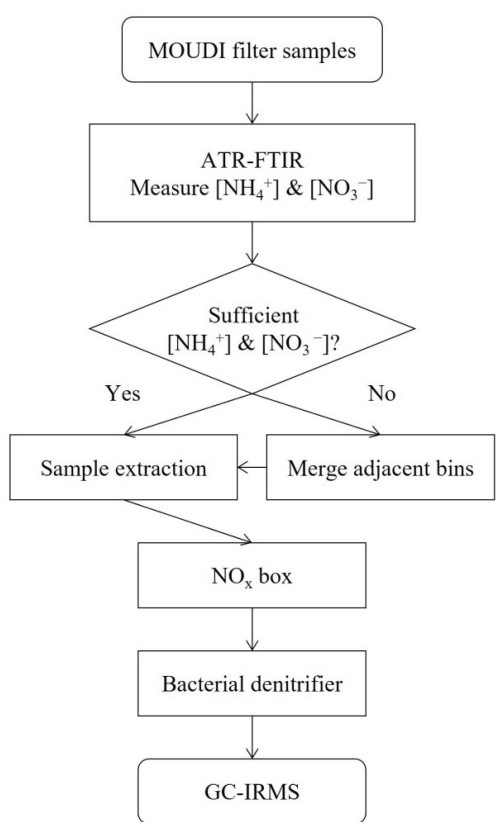

**Figure 1. Schematic diagram of sampling and isotope analysis procedures.**

Subsequently, the bacterial denitrifier method was employed to measure the $\delta^{15}$N of TN, as well as both
$\delta^{15}$N and $\delta^{18}$O of nitrate + nitrite ($NO_3^- + NO_2^-$, NN). Two bacterial strains were employed: *Pseudomonas
chlororaphis* (ATCC® 43928™, Manassas, VA, USA) for TN analysis and *Pseudomonas chlororaphis ssp.
aureofaciens* (ATCC® 13985™, Manassas, VA, USA) for NN analysis. These bacteria converted sample $NO_3^-$ to
nitrous oxide ($N_2O$) while preserving the original isotopic compositions of both N and O elements (Weigand et al.,
2016). Gas chromatography-isotope ratio mass spectrometry (GC-IRMS) was used to analyze the resulting $N_2O$ for
the simultaneous determination of $\delta^{15}$N and $\delta^{18}$O values. The system was calibrated using two international isotope
standards: USGS 34 ($\delta^{15}$N = −1.8‰; $\delta^{18}$O = −27.93‰) and IAEA-NO$_3$ ($\delta^{15}$N = +4.7‰; $\delta^{18}$O = +25.61‰). Detailed
descriptions of isotope measurement procedures are provided by Chen, T. Y. et al. (2022).

Isotopic values measured for NN were considered representative of $NO_3^-$ (*i.e.*, $\delta^{15}$N-NN $\approx$ $\delta^{15}$N-NO$_3^-$;
$\delta^{18}$O-NN $\approx$ $\delta^{18}$O-NO$_3^-$) due to negligible $NO_2^-$ concentrations detected by IC analysis. Since TN primarily
comprises $NH_4^+$ and NN, the isotopic values measured for TN minus those for NN were assumed to represent $NH_4^+$,
expressed as:

$$\delta^{15}\text{N-NH}_4^+ = \frac{\delta^{15}\text{N}_{\text{TN}} \times M_{\text{TN}} - \delta^{15}\text{N}_{\text{NN}} \times M_{\text{NN}}}{M_{\text{TN}} - M_{\text{NN}}}, \tag{1}$$





where $M_{TN}$ and $M_{NN}$ represent the molarities of TN and NN in the sample solution, respectively, measured by the photolytic NO/NO$_2$/NO$_x$ analyzer (NO$_x$ box, Model T200P, Teledyne API). The NH$_4^+$ concentration was determined on a fluorescence spectrophotometer (Hitachi F-2700) using a fluorometric method (Holmes et al., 1999). For quality control, $\delta^{15}$N-NH$_4^+$ isotope data points were excluded from analysis if NN comprised greater than 80% of TN or if NH$_4^+$ represented less than 60% of TN−NN. The latter criterion was based on the assumption that contributions from organic nitrogen were negligible under these conditions.

### 2.3. Bayesian isotope mixing model

Source contributions of NH$_3$ and the formation pathways of $p$NO$_3^-$ were estimated using the MixSIAR framework (v3.1.12), which employs Bayesian models to estimate source contributions while accounting for uncertainties in source values (Stock et al., 2018). The Bayesian model is formulated as:

$$X_{ij} = \sum_{k=1}^{K} P_k \times S_{jk} + \varepsilon_{ij} \tag{2}$$

where $X_{ij}$ is the isotope value of tracer $j$ ($\delta^{15}$N-NH$_4^+$ or $\delta^{18}$O-NO$_3^-$) for the merged filter sample $i$, $P_k$ is the proportion of source $k$ with $\sum P_k = 1$, $S_{jk}$ is the isotope value of tracer $j$ for source $k$, following a normal distribution with mean values and standard deviations, and $\varepsilon_{ij}$ is the residual error. The model was run with extended Markov Chain Monte Carlo (MCMC) parameters to ensure convergence, with all runs confirmed to have converged based on the Gelman-Rubin potential scale reduction factor and the Geweke diagnostic (Stock and Semmens, 2016). Results are reported as mean source contributions with associated standard deviations. To further assess model performance, reconstructed $X_{ij}$ was calculated by weighting $S_{jk}$ with the model-derived $P_k$ and compared to the measured $X_{ij}$.

### 2.3.1. Source apportionments of NH$_3$

Determining NH$_3$ emission sources requires analyzing the emitted $\delta^{15}$N-NH$_3$ values. However, due to isotopic fractionation during the conversion of NH$_3$ to $p$NH$_4^+$, the measured $\delta^{15}$N-NH$_4^+$ values do not directly reflect NH$_3$ sources. Since direct $\delta^{15}$N-NH$_3$ measurements were not available in this study, the initial $\delta^{15}$N-NH$_3$ ($\delta^{15}$N-NH$_3^0$) was estimated from $\delta^{15}$N-NH$_4^+$ using an empirical relationship (Pan et al., 2016):

$$\delta^{15}\text{N-NH}_3{}^0 = \delta^{15}\text{N-NH}_4^+ - \varepsilon_{\text{NH}_4^+-\text{NH}_3}\left(1-f\right) \tag{3}$$

where $\delta^{15}$N-NH$_4^+$ represents the concentration-weighted mean $\delta^{15}$N-NH$_4^+$ at Xitou, $\varepsilon_{\text{NH}_4^+-\text{NH}_3}$ is the isotope fractionation constant assumed as +33‰ (Heaton et al., 1997), and $f$ is the fraction of $p$NH$_4^+$ in the NH$_3$−$p$NH$_4^+$ system. The $f$ values were estimated using the Community Multiscale Air Quality (CMAQ) model (version 4.7.1), incorporating meteorological data from the Weather Research and Forecasting (WRF) model (version 3.7.1) and the emission database from the Taiwan Emission Data System (version 12) (Tsai et al., 2024). The model-simulated $p$NH$_4^+$ concentrations agreed with laboratory measurements within a 20% uncertainty range (Fig. S2).

For NH$_3$ source apportionment, we considered four major emission sources, each with characteristic $\delta^{15}$N-NH$_3$ values: fertilizer (−28.3±5.8‰), waste (−17.6±5.6‰), NH$_3$ slip (−8.2±5.5‰), and fossil fuel emissions (1.8±3.2‰) (Kawashima et al., 2023). These sources can be categorized into two groups based on their $\delta^{15}$N-NH$_3$ signatures: (1) volatilization-related sources (fertilizer and waste) with lower $\delta^{15}$N-NH$_3$ values and (2) combustion-related sources (NH$_3$ slip and fossil fuel emissions) with higher values (Chen, Z.-L. et al., 2022). $\delta^{15}$N-NH$_3$ obtained using passive techniques was adjusted by 15‰ to account for systematic differences between collection methods (Kawashima et al., 2023; Walters et al., 2020). Statistical values for MixSIAR analysis are provided in Table S1.

### 2.3.2. Estimating formation pathways of $p$NO$_3^-$

To evaluate the contribution of different HNO$_3$ formation pathways, six potential reaction mechanisms were considered, each characterized by $\delta^{18}$O-NO$_3^-$ signatures, as shown in Fig. S3. These pathways were classified based



on the initial reaction step: PXa for the formation initiated by the reaction of NO with $RO_2$, while PXb for the reaction of $NO_2$ and $O_3$. The variable X (1, 2, or 3) denotes the oxidants involved in the second reaction step: OH (from $H_2O$) as P1, OH (from $O[^1D] + H_2O$) as P2, and $O_3$ as P3. The $\delta^{18}O$ values of key oxidants were assigned based on their atmospheric sources. $RO_2$ is assumed to be 23.5‰, reflecting that O in $RO_2$ originates from molecular $O_2$ (Kroopnick and Craig, 1972). $O_3$ exhibits elevated $\delta^{18}O$ values ranging from 90 to 122‰ (Hastings et al., 2003). OH radicals exhibit different $\delta^{18}O$ signatures depending on their formation pathway. A lower $\delta^{18}O$ signature ($-15$ to 0‰) for OH directly derived from the oxygen atom in $H_2O$ (Dubey et al., 1997), and a higher $\delta^{18}O$ (38 to 61‰) for OH formed via the reaction of $O(^1D)$ (produced from $O_3$ photolysis) with $H_2O$ due to the oxygen contribution from $O_3$. The $\delta^{18}O$ of $HNO_3$ was calculated using a mass-balance approach, assuming no kinetic isotope fractionation (Walters and Michalski, 2016). For daytime formation, four pathways were considered: P1a ($\delta^{18}O\text{-}NO_3^-$: 33−49‰), P2a ($\delta^{18}O\text{-}NO_3^-$: 50−69‰), P1b ($\delta^{18}O\text{-}NO_3^-$: 55−81‰), and P2b ($\delta^{18}O\text{-}NO_3^-$: 73−102‰). For nighttime formation, pathways P3a ($\delta^{18}O\text{-}NO_3^-$: 50−69‰) and P3b ($\delta^{18}O\text{-}NO_3^-$: 73−102‰) were considered, while retaining P1a to account for residual from daytime processes (Fig. S3). The inclusion of P1a in nighttime analysis is necessary because the observed nighttime $\delta^{18}O\text{-}NO_3^-$ values could not be fully explained by typical nocturnal formation pathways (*i.e.*, P3a and P3b) alone. To simplify the nighttime analysis, P2a, P1b, and P2b were excluded due to their overlapping $\delta^{18}O$ signatures with P3a and P3b. However, it is possible that the estimated fractions of P3a and P3b include partial contributions from these excluded pathways.

## 3. Results and discussion

### 3.1. Environmental variabilities

Figure 2 shows various environmental parameters, including temperature (Temp), relative humidity (RH), wind speed (WS), wind direction (WD), radiation (Rad), visibility (Vis), and carbon monoxide (CO) concentration monitored during the sampling period from 17 to 24 April 2021. During the clear period, distinct diurnal variations were observed in temperature, RH, wind direction, solar radiation, and CO concentrations. Daytime temperatures were higher ($20\pm2$ °C) than nighttime ($15\pm1$ °C), while RH exhibited the opposite trend, increasing to $99\pm1$ % at night compared to $87\pm8$ % during daytime. Wind patterns followed typical valley-mountain circulation, with daytime valley winds predominant from the north (316°−33°) and nighttime mountain winds shifted to the southeast (124°−179°), consistent with previous observations in Xitou (Chen et al., 2021). The combined effects of daytime valley winds and sea breezes facilitated the daily transport of air masses from upstream urban areas toward the downstream mountain site, likely introducing anthropogenic pollutants. This transport pattern is corroborated by CO concentrations, a tracer of combustion emissions, which were consistently higher during the daytime ($0.23\pm0.03$ ppm) than nighttime ($0.14\pm0.03$ ppm). However, meteorological observations during the sampling period revealed a prolonged fog that disrupted these typical diurnal patterns and influenced local atmospheric dynamics and Nr behavior. This prolonged fog occurred from 18 April 12:00 to 19 April 14:00, identified by visibility below 1000 m and RH above 90% for at least one hour. During this ~26-hour period, wind speed decreased from $0.9\pm0.6$ m s$^{-1}$ to $0.5\pm0.5$ m s$^{-1}$, with approximately 25% of observations recording <0.1 m s$^{-1}$, indicating stagnant conditions. Meanwhile, temperature ($15\pm1$ °C) and RH ($99.5\pm0.2$ %) remained relatively stable, and CO concentrations ($0.22\pm0.03$ ppm) were comparable to those observed during clear daytime, indicating that pollutants were not removed during nighttime and that transport was limited, creating a closed-system environment conducive to in-situ chemical transformations.

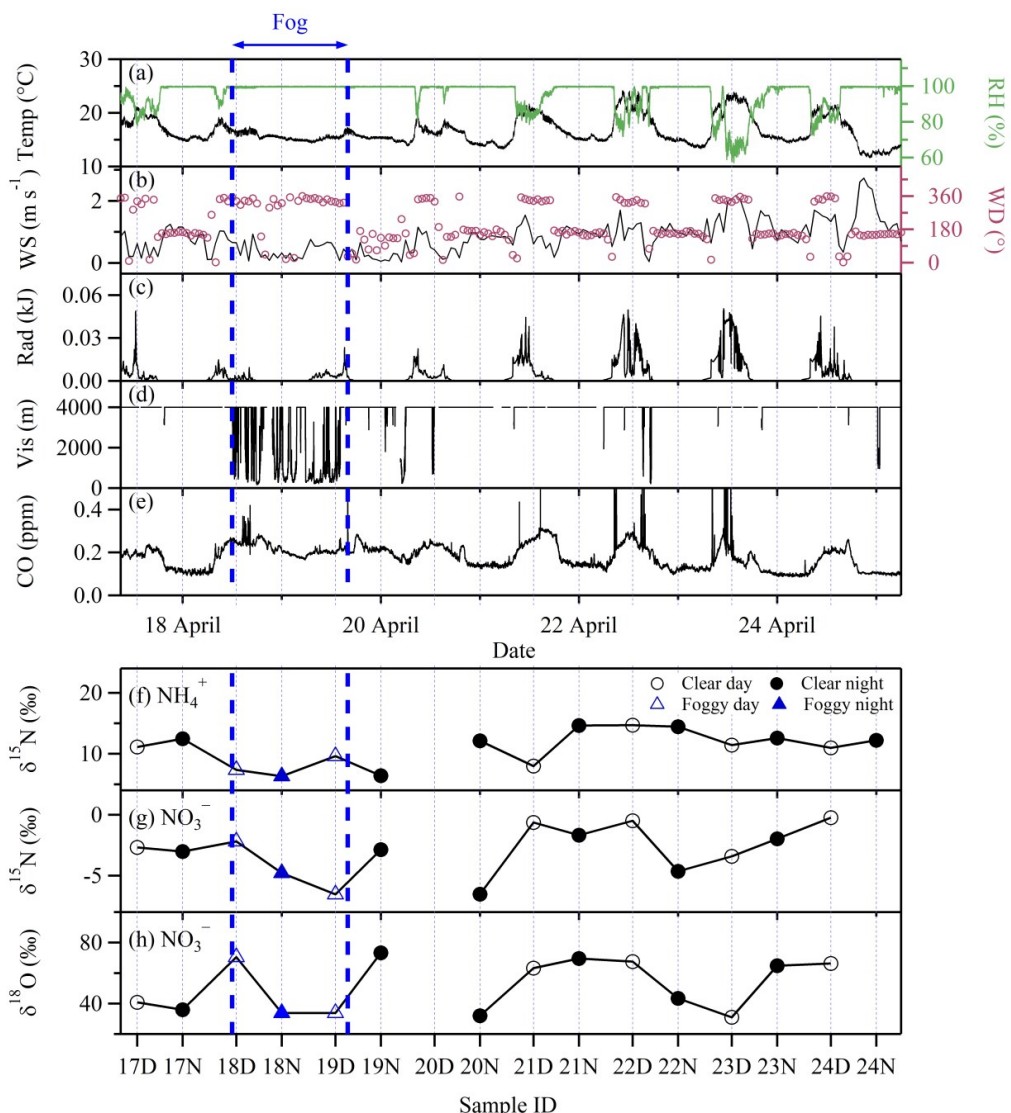

**Figure 2. Time series of various environmental parameters measured from 17−24 April 2021, including (a) temperature (Temp) and relative humidity (RH), (b) wind speed (WS) and wind direction (WD), (c) radiation (Rad), (d) visibility (Vis), and (e) CO concentration. The concentration-weighted isotope values for (f) $\delta^{15}$N-NH$_4^+$, (g) $\delta^{15}$N-NO$_3^-$, and (h) $\delta^{18}$O-NO$_3^-$ over daytime (09:00−17:00 LT, D) and nighttime (18:00−06:00 LT the next day, N) sampling periods.**

### 3.2. Size-resolved aerosol chemical composition

The effects of the prolonged fog on the mass concentration distributions of NH$_4^+$, NO$_3^-$, SO$_4^{2-}$, and black carbon (BC) as a function of particle size are shown in Fig. 3. During the clear period (Fig. 3, left panels), these four species exhibited higher daytime concentrations than nighttime values. NH$_4^+$, SO$_4^{2-}$, and BC exhibited peak





concentrations in the 0.32−0.56 μm range during the daytime, and shifted to 1−1.8 μm with a broader distribution at nighttime. In comparison, $NO_3^-$ showed a bimodal distribution, with peaks at 0.56−1 μm and 3.2−5.6 μm during daytime, shifting to 1−1.8 μm and 3.2−5.6 μm at nighttime. These variations likely reflect coagulation processes contributing to particle growth under elevated RH conditions. During the foggy period (Fig. 3, right panels), the mass concentration distributions on 18D remained within the variability range observed during clear periods, with peaks around 0.32−1 μm. Compared with clear days, nitrate concentrations in the sub-micrometer region increased on 18D, surpassing those observed in the coarse mode. This enhancement likely resulted from enhanced $HNO_3$-$NH_3$ partitioning processes, promoted by the high surface-to-volume ratio of sub-micrometer particles during the hygroscopic growth at high RH. On 18N, all species exhibited peak shifts to 1−1.8 μm, indicating hygroscopic growth. On the following day (19D), concentrations decreased, likely resulting from wet deposition of larger droplets and reduced pollutant transport under stagnant conditions. A time series showing the changes in concentration of $pNH_4^+$, $pNO_3^-$, $pSO_4^{2-}$, and BC is provided in Fig. S4. These observations demonstrate that prolonged fog conditions enhanced local transformation processes and altered particle size distributions. The changes in particle size and composition laid the foundation for interpreting isotopic shifts and reactive nitrogen transformations in subsequent sections.

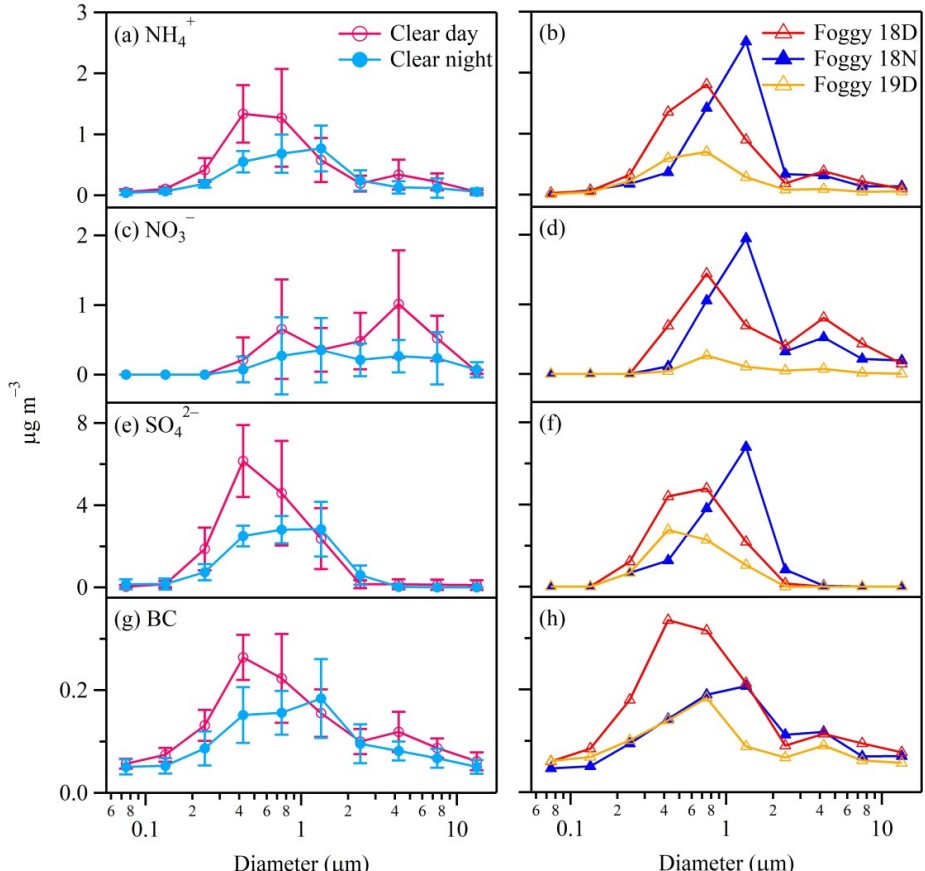

**Figure 3. Size-resolved mass concentration distributions of particulate (a,b) $NH_4^+$, (c,d) $NO_3^-$, (e,f) $SO_4^{2-}$, and (g,h) black carbon (BC) estimated from FTIR analysis during clear (left panels) and foggy (right panels) periods.**





### 3.3. $\delta^{15}N$-$NH_4^+$ and derivation of emitted $\delta^{15}N$-$NH_3$

### 3.3.1. $\delta^{15}N$-$NH_4^+$

The daily concentration-weighted $\delta^{15}N$-$NH_4^+$ values at the Xitou site ranged from 6.31‰ to 14.69‰, with an average of 10.95 ± 2.76‰ (Fig. 2f, calculation details in Supplementary Description S3). This result is consistent with previous winter observations at the same site (Chen, T. Y. et al., 2022) and falls within the range reported for both urban and remote locations, as shown in Fig. S5 (Chen, T. Y. et al., 2022; Kawashima et al., 2023; Hall et al., 2016; Kundu et al., 2010; Moore, 1977; Proemse et al., 2012; Savard et al., 2017; Ti et al., 2018; Walters et al., 2022). Temporal patterns showed higher $\delta^{15}N$-$NH_4^+$ during the clear period, with daytime and nighttime averages of

11.22 ± 2.13‰ and 12.12 ± 2.53‰, respectively (Table 1). In contrast, lower values were often observed during fog, decreasing from 7.35‰ (18D) to 6.31‰ (18N), before increasing to 9.60‰ (19D). The decrease may result from the absence of fresh $NH_3$ input under stagnant fog conditions, allowing continuous isotopic fractionation as $NH_3$ converts to $p NH_4^+$ (Walters et al., 2019). The observed decrease in $\delta^{15}N$-$NH_4^+$ during the foggy period contrasts with our previous study, which reported a slight increase in $\delta^{15}N$-$NH_4^+$ under foggy conditions (Chen, T. Y. et al., 2022).

This discrepancy likely reflects that previous fog events, lasting only ~6 hours, were insufficient to permit detectable isotopic fractionation processes (Chen, T. Y. et al., 2022; Chen et al., 2021).

**Table 1.** Concentration-weighted isotope values (unit: ‰) under different weather circumstances.

|  | $\delta^{15}N$-$NH_4^+$ | $\delta^{15}N$-$NO_3^-$ | $\delta^{18}O$-$NO_3^-$ |
|---|---|---|---|
| All | 10.81±2.72 (n = 15) | −2.98±1.97 (n = 14) | 51.82±16.47 (n = 14) |
| Clear day | 11.22±2.13 (n = 5) | −1.50±1.30 (n = 5) | 53.74±14.99 (n = 5) |
| Clear night | 12.12±2.53 (n = 7) | −3.46±1.67 (n = 6) | 53.12±16.59 (n = 6) |
| Foggy 18D | 7.35 (n = 1) | −2.19 (n = 1) | 70.44 (n = 1) |
| Foggy 18N | 6.31 (n = 1) | −4.80 (n = 1) | 33.81 (n = 1) |
| Foggy 19D | 9.60 (n = 1) | −6.54 (n = 1) | 33.81 (n = 1) |

The size-resolved $\delta^{15}N$-$NH_4^+$ patterns shown in Fig. 4 revealed an increase from 7.14‰ to 14.59‰ with particle diameter up to ~1 μm before a decline from 16.08‰ to 7.53‰ in the coarse mode (> 1 μm), reflecting differences in particle formation and transformation processes. We hypothesize that fine particles with a diameter less than 1 μm were formed near the sampling site, as their lower $\delta^{15}N$-$NH_4^+$ values are consistent with emissions from biogenic sources. On the other hand, coarse particles (1−10 μm, $PM_{1-10}$) are likely produced from urban areas,

where $HNO_3$ reacts with sea salt or dust to form nitrate-rich aerosols that were subsequently transported to the sampling site. Under high RH and near-neutral pH conditions (Fig. S6), isotopic exchange between $NH_4^+$ in coarse-mode particles and locally available $NH_3$ with lower $\delta^{15}N$ may occur via aqueous-phase reactions, resulting in isotopic depletion in the coarse fraction. During the foggy period, the bell-shaped pattern became less apparent, with $\delta^{15}N$-$NH_4^+$ values remaining relatively uniform across particle sizes. This was likely due to suppressed gas-particle

exchange under stagnant and high RH conditions, combined with elevated $NH_4^+$ concentrations (Fig. 3) that promoted rapid $HNO_3$ uptake and neutralization in the particle phase, shifting the system toward equilibrium (Walters et al., 2019; Chen, T. Y. et al., 2022). A similar uniform distribution pattern was observed on 19N and 21D, which also exhibited comparably low $\delta^{15}N$-$NH_4^+$ values (Fig. 2f).




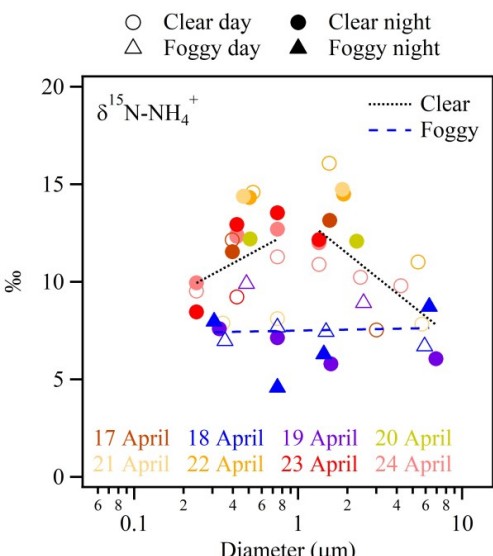

**Figure 4. Size-resolved isotopic composition δ$^{15}$N of $p$NH$_4^+$ with a dotted (dashed) line indicating the linear regression trend during the clear (foggy) period.**

### 3.3.2. δ$^{15}$N-NH$_3^0$ and its source apportionment

The derived δ$^{15}$N-NH$_3^0$ ranged from −12.55‰ to −6.71‰ and was consistently lower than the concentration-weighted δ$^{15}$N-NH$_4^+$, as shown in Fig. 5a. Based on the derived δ$^{15}$N-NH$_3^0$ values and assuming a single-source contribution, NH$_3$ can be mainly attributed to NH$_3$ slip. However, in real atmospheric conditions, air parcels are typically influenced by a mixture of multiple sources rather than a single dominant contributor. To quantitatively assess the contributions of different NH$_3$ sources to the observed particulate NH$_4^+$, the MixSIAR framework was employed, with the results summarized in Fig. 5b. Overall, the two combustion-related sources (fossil fuel and NH$_3$ slip) were the dominant contributors to NH$_3$ in the Xitou area, accounting for 50−83% of the total NH$_3$ emissions. During periods with higher δ$^{15}$N-NH$_3^0$, such as 17D, 17N, and 19D, fossil fuel combustion contributed an average of 54±3% to NH$_3$, followed by NH$_3$ slip (26±1%), waste (13±1%), and fertilizer (8±1%). In comparison, during periods with lower δ$^{15}$N-NH$_3$, the contributions were more evenly distributed: fossil fuel combustion and NH$_3$ slip contributed 30±5% and 29±1%, respectively, while waste and fertilizer sources increased to 24±2% and 17±3% (Fig. S8). The dominance of combustion-related sources at the remote mountain site, even during the stagnant fog condition, demonstrates the influence of long-range transport from urban and industrial areas. These findings underscore the need for regional emission control strategies targeting anthropogenic NH$_3$ sources, which represent the most efficient and cost-effective mitigation strategy in Taiwan compared to NO$_x$ or SO$_2$ reduction (Huang, P. C. et al., 2024).



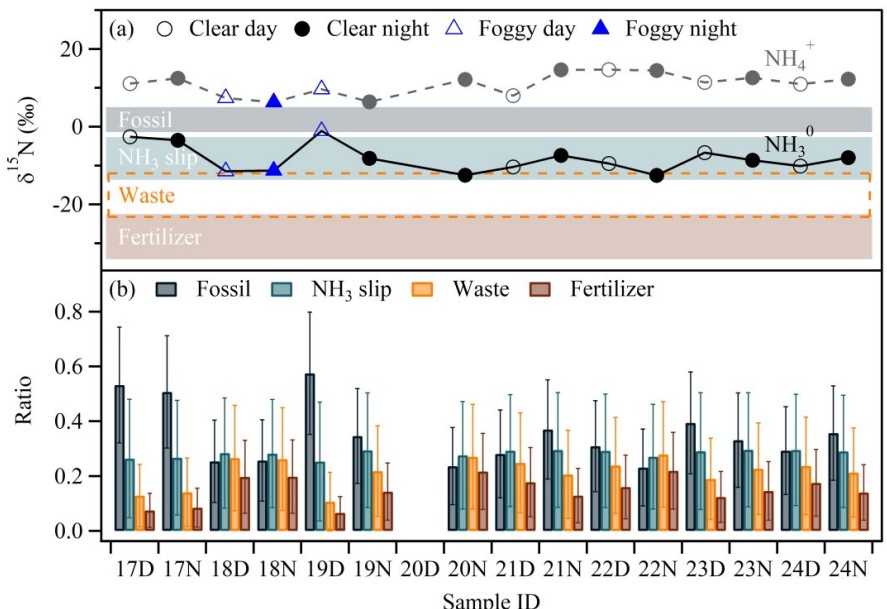

**Figure 5. (a) Estimated δ¹⁵N-NH₃⁰ at Xitou derived from measured δ¹⁵N-NH₄⁺, and compared with the characteristic δ¹⁵N ranges of NH₃ sources reported in the literature (Table S1). (b) Source apportionment results of NH₃ from the MixSIAR framework.**

**3.4. $\delta^{15}$N and $\delta^{18}$O of $p$NO₃⁻**

**3.4.1. $\delta^{15}$N-NO₃⁻**

The daily concentration-weighted $\delta^{15}$N-NO₃⁻ values ranged from −6.54‰ to −0.24‰, with an average of −2.98±1.97‰ (Fig. 2g). These values are consistent with those reported in mountain regions such as Mt. Lulin (−3.6±3.8‰) (Guha et al., 2017) and the Himalayan-Tibetan Plateau (0.44±4.89‰) (Lin et al., 2021), as illustrated

in Fig. S9. Compared to wintertime measurements at the same site (2.98±1.20‰) (Chen, T. Y. et al., 2022), the lower springtime values likely reflect seasonal shifts in NO$_x$ sources and atmospheric processing. They are also significantly lower than typical urban values, such as ~11.5‰ in Beijing and Gosan (Fan et al., 2020; Kundu et al., 2010), indicating that long-range atmospheric transport from urban sources to the mountain site is accompanied by progressive isotopic fractionation during NO$_x$-to-HNO₃ conversion, resulting in depletion of $\delta^{15}$N-NO₃⁻ values in

mountain regions (Freyer et al., 1993; Gobel et al., 2013).

Under clear conditions, $\delta^{15}$N-NO₃⁻ showed minor diurnal variation, with slightly higher values during daytime (−1.50±1.30‰) than nighttime (−3.46±1.67‰; Table 1), reflecting typical shifts between transported urban and local biogenic NO$_x$ sources. During the fog, $\delta^{15}$N-NO₃⁻ decreased from −2.19‰ (18D) to −4.80‰ (18N), and further to −6.54‰ (19D) (Fig. 2g), indicating a rapid shift in formation processes and nitrate sources. As observed

for NH₄⁺, this decline differs from the previous study due to the extended fog duration (Chen, T. Y. et al., 2022). It reflects the combined effects of isotopic partitioning between NO and NO₂, isotope fractionation during NO$_x$-to-$p$NO₃⁻ conversion, and reduced atmospheric transport, which limited the influx of urban air masses (Vicars et al., 2013; Gobel et al., 2013). Weak wind speeds further enhanced $p$NO₃⁻ formation in mountainous regions under high RH, where aqueous-phase reactions dominated. Simultaneously, the stagnation of air masses likely increased the

relative contribution of local biogenic NO$_x$ sources, which are typically more depleted in $\delta^{15}$N.





Size-segregated $\delta^{15}N$-$NO_3^-$ values ranged from −10.05 to 0.78‰, showing a weak positive correlation with particle diameter (Fig. 6a). Fine particles (PM$_1$) had slightly lower $\delta^{15}N$-$NO_3^-$ values (−3.59±2.38‰) compared to larger particles (PM$_{1-10}$, −2.07±2.10‰), consistent with our previous findings (Chen, T. Y. et al., 2022). This pattern likely reflects isotopic fractionation during HNO$_3$ formation and subsequent partitioning across size modes. In urban areas near major NO$_x$ sources, HNO$_3$ might react with coarse-mode particles (*e.g.*, NaCl or dust), forming $\delta^{15}N$-enriched $p$NO$_3^-$. The remaining NO$_x$, which is depleted in $\delta^{15}N$, subsequently forms HNO$_3$ that condenses onto fine particles formed near the sampling site (Gobel et al., 2013). Additionally, fine-mode $p$NO$_3^-$ typically forms through gas-phase oxidation of NO$_x$ followed by HNO$_3$ condensation under sufficient NH$_3$, preserving the $\delta^{15}N$-depleted signature of precursor gases. In contrast, coarse-mode $p$NO$_3^-$ often forms through heterogeneous reactions of NO$_2$ or HNO$_3$ on particle surfaces near the emission sources, which preferentially enrich heavier isotopes.

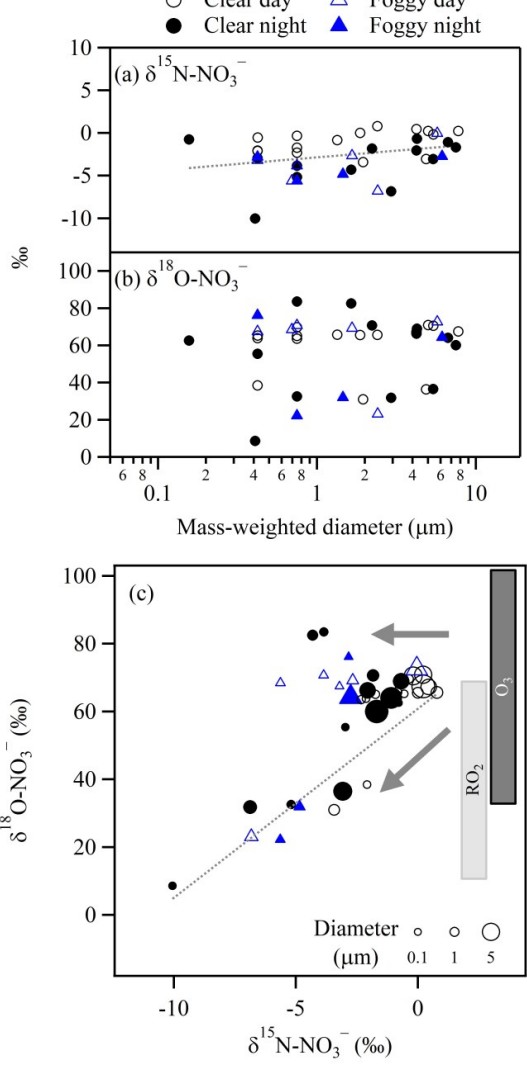

**Figure 6. Size-resolved isotopic composition of $p$NO$_3^-$. (a) $\delta^{15}N$-$NO_3^-$ values as a function of mass-weighted particle diameter, with a dotted line indicating the linear regression trend. (b) $\delta^{18}O$-$NO_3^-$ values as a function**





of mass-weighted particle diameter. (c) Two-dimensional plot of $\delta^{15}N$ versus $\delta^{18}O$ across particle sizes. The dotted line provides a visual guide to possible source groupings. The gray boxes labeled "RO$_2$" and "O$_3$" indicate the potential $\delta^{18}O$-NO$_3^-$ ranges resulting from reactions with RO$_2$ and O$_3$, respectively.

### 3.4.2. $\delta^{18}O$-NO$_3^-$

The daily concentration-weighted $\delta^{18}O$-NO$_3^-$ ranged from 30.98 to 73.27‰, with an average value of 51.82±16.47‰ (Fig. 2h). These values are lower than our previous winter observations (72.66±3.42‰) (Chen, T. Y. et al., 2022), but align with measurements from other mountain regions such as 10.8−92.4‰ at Mt. Lulin (Guha et al., 2017) and 64.71±11.52‰ at the Himalayan-Tibetan Plateau (Lin et al., 2021), as illustrated in Fig. S10. A sharp decrease in $\delta^{18}O$-NO$_3^-$ from 70.44‰ (18D) to 33.81‰ (18N and 19D) occurred during the fog, coinciding with $\delta^{15}N$-NO$_3^-$ depletion. These declines indicate a shift from O$_3$- to RO$_2$-dominated oxidation, likely driven by stagnant winds, suppressed urban input, and enhanced local aqueous-phase chemistry under high RH.

A fair correlation ($r$ = 0.66) between $\delta^{18}O$-NO$_3^-$ and $\delta^{15}N$-NO$_3^-$ highlights the interplay between oxidation processes and nitrogen cycling. As shown in Fig. 6c, two distinct regimes emerge: particles with higher $\delta^{18}O$-NO$_3^-$ (55−83‰) exhibit enriched $\delta^{15}N$-NO$_3^-$ (−6 to 1‰), while those with lower $\delta^{18}O$-NO$_3^-$ (9−38‰) tend to have depleted $\delta^{15}N$-NO$_3^-$ (−10 to −2‰). Considering the evolution of air parcels from urban to rural regions, particles with the highest $\delta^{15}N$-NO$_3^-$ (~0‰) and $\delta^{18}O$-NO$_3^-$ (~70‰) likely form in metropolitan areas, where freshly emitted NO contributes to elevated $\delta^{15}N$-NO$_3^-$ (Gobel et al., 2013), and O$_3$-driven oxidation (pathways PXb, Fig. S3) leads to higher $\delta^{18}O$-NO$_3^-$. During atmospheric transport from urban to mountainous areas, $\delta^{15}N$-NO$_3^-$ values gradually decrease from ~0‰ to −6‰ due to isotopic fractionation (Gobel et al., 2013), while $\delta^{18}O$-NO$_3^-$ values remain elevated (~60−80‰) because O$_3$-driven oxidation continues to dominate in the O$_3$-rich urban plumes. This intermediate regime represents particles formed during the transport process, maintaining the O$_3$ signature while experiencing nitrogen isotopic depletion. On the other hand, particles with the lowest $\delta^{15}N$-NO$_3^-$ (−10 to −2‰) and $\delta^{18}O$-NO$_3^-$ (9−38‰) likely form in rural mountainous regions, where oxidation pathways involving RO$_2$ (pathways PXa, Fig. S3) result in lower $\delta^{18}O$-NO$_3^-$. The lower $\delta^{15}N$-NO$_3^-$ values may result from continued isotopic depletion during NO$_x$ transport and the increased contributions from $\delta^{15}N$-depleted biogenic emissions, especially under stagnant fog conditions.

The daily $\delta^{18}O$-NO$_3^-$ and $\delta^{15}N$-NO$_3^-$ relationship depicted in Fig. S11 and Fig. S12 further illustrates this evolution, exhibiting a shift toward lower isotopic values during the fog period, then increasing to higher values as transport resumed. The observed positive correlation between $\delta^{15}N$-NO$_3^-$ and $\delta^{18}O$-NO$_3^-$ is consistent with previous studies (Chang et al., 2018; Chen, T. Y. et al., 2022; Fan et al., 2020; Guha et al., 2017; Hall et al., 2016; Lin et al., 2021; Proemse et al., 2012; Savard et al., 2017; Vicars et al., 2013; Zhao et al., 2020), which have shown that higher $\delta^{15}N$-NO$_3^-$ and $\delta^{18}O$-NO$_3^-$ values are typically associated with polluted urban environments, while lower values are more common in less polluted regions (Fig. S13). Finer size-resolved measurements can reveal the evolution of nitrate formation as air masses move from emission source regions to rural sites. A quantitative analysis of $p$NO$_3^-$ formation is provided in the following section.

### 3.4.3. Contributions of $p$NO$_3^-$ formation pathways

Formation pathways of $p$NO$_3^-$ were quantitatively analyzed using the MixSIAR framework based on $\delta^{18}O$-NO$_3^-$ values as shown in Fig. S14a. In the daytime, the RO$_2$-initiated P1a (RO$_2$ + NO → NO$_2$, followed by NO$_2$ + ·OH → HNO$_3$, Fig. S3) accounted for over 82−92% of $p$NO$_3^-$ formation on 17D, 19D, and 23D (Fig. S14a and b). These periods were characterized by low $\delta^{18}O$-NO$_3^-$ values (31−41‰) and low CO concentrations (< 0.22ppm, Fig. S15), suggesting limited influence from urban air masses and the dominance of local RO$_2$ oxidation processes. In contrast, during daytime with higher $\delta^{18}O$-NO$_3^-$ values (63−70‰), such as 18D, 21D, 22D, and 24D, P2a, P1b, and P2b were the dominant formation pathways, contributing an average of 29±1%, 27±3% and 28±5%, respectively. Except for 24D where CO concentration was 0.20 ppm, these days corresponded to elevated CO concentrations (0.24−0.27 ppm, Fig. S15), suggesting enhanced transport of urban pollutants.



At night, the formation of $p$NO$_3^-$ became more complex. P1a remained a major contributor, which accounted for 80−94% of $p$NO$_3^-$ formation on several nights (17N, 18N, 20N, and 22N) when $\delta^{18}$O-NO$_3^-$ values were lower (32−43‰). The presence of P1a during nighttime likely resulted from residual ·OH produced during the daytime or from non-photolytic ·OH generation via reactions between O$_3$ and alkenes or terpenes (Kroll et al., 2001; Aschmann et al., 2002). During the periods with elevated $\delta^{18}$O-NO$_3^-$ values (19N, 21N, 23N, $\delta^{18}$O-NO$_3^-$ values

ranging from 65 to 73‰), pathways P3a and P3b became the dominant contributors, accounting for 33±6% and 47±9% of $p$NO$_3^-$ formation, respectively. The complexity in formation pathways, as well as weaker wind speed and relatively lower CO concentration, may lead to a weaker correlation between $\delta^{18}$O-NO$_3^-$ values and CO concentrations during nighttime.

      During the fog, a shift in nitrate formation was observed. On 18D, consistent with high $\delta^{18}$O-NO$_3^-$ values

(70‰) and elevated pollutant concentrations (CO: 0.24 ppm), the dominant formation pathways were P2a (30±20%) and P2b (36±16%). As fog intensified during 18N and persisted into 19D, $\delta^{18}$O-NO$_3^-$ values decreased sharply to 34‰, and the dominant pathway shifted to P1a (93±5% on 18N, and 90±5% on 19D). This transition reflects the suppression of long-range transport and O$_3$ photochemistry by the fog, leading to conditions favorable for local RO$_2$-driven oxidation.

**4. Conclusions**

      Our observations in a subtropical mountain forest in Taiwan revealed evident diurnal and meteorological influences on the transport and transformation of reactive nitrogen (Nr). Daytime particle concentrations consistently exceeded nighttime levels, reflecting upslope transport of urban plumes, while a 26-hour fog suppressed urban input and enhanced local transformation processes. During the prolonged fog, progressive decreases in $\delta^{15}$N-

NH$_4^+$ and $\delta^{15}$N-NO$_3^-$ values indicated continued isotopic fractionation under stagnant conditions. Simultaneously, $\delta^{18}$O-NO$_3^-$ values shifted markedly, suggesting a transition from O$_3$- to RO$_2$-dominated $p$NO$_3^-$ formation pathways. Unlike previous studies of shorter fog episodes, this extended fog allowed more precise identification of isotopic fractionation dynamics and nitrate evolution pathways. Size-resolved $\delta^{15}$N-NO$_3^-$ and $\delta^{18}$O-NO$_3^-$ patterns further revealed two distinct $p$NO$_3^-$ formation regimes: one associated with long-range transport from urban plumes under

O$_3$ oxidation, and another linked to local RO$_2$-driven processes with greater isotopic depletion and biogenic contributions.

      Bayesian source apportionment based on $\delta^{15}$N-NH$_3$ constraints indicated that combustion-related emissions contributed 50–83% of the total NH$_3$, even under stagnant, foggy periods when fresh urban inputs were limited. This persistent anthropogenic signature highlights the resilience of combustion sources in shaping local nitrogen budgets.

Isotopic analysis of $\delta^{18}$O-NO$_3^-$ further revealed that RO$_2$-initiated oxidation accounted for 42–95% of daytime $p$NO$_3^-$ formation and remained as a substantial contributor at night, when heterogeneous reactions accounted for 6–84% of total $p$NO$_3^-$ formation. These findings highlight the significance of RO$_2$ chemistry in nitrate formation, particularly in forested, high-humidity environments where biogenic VOC emissions promote RO$_2$ radical production.

Despite these insights, certain methodological uncertainties remain. Estimation of $\delta^{15}$N-NH$_3$ from particulate $\delta^{15}$N-NH$_4^+$ relied on simplified equilibrium fractionation assumptions that may underestimate kinetic effects during gas–particle conversion, potentially skewing source attribution toward combustion-related NH$_3$ sources. Similarly, the overlapping isotopic signatures of NO$_x$ to $p$NO$_3^-$ complicate efforts to isolate individual oxidation processes. Future studies should integrate direct gas-phase isotope observations, chamber studies of

aqueous-phase reactions, and region-specific source inventories to refine isotopic models. Such advancements will improve our ability to distinguish urban versus biogenic contributions, assess the role of local photochemistry, and evaluate the effectiveness of targeted emission reduction strategies.



**Code availability.** Codes are available upon request.


**Data availability.** Data are available upon request.

**Sample availability.** Samples are no longer available due to full consumption during analytical procedures.

**Author contribution.** HMH designed the project. WCH conducted the field observations and collected filter samples. MHH and WCH carried out the laboratory experiments. MHH, WCH, and WCL contributed to data curation. MHH and WCL performed the formal analysis. JPC conducted the CMAQ modeling and analysis. YJL collected the meteorological data. HR supervised the isotope measurements and analysis. WCL performed MixSIAR analysis, prepared the visualizations, and wrote the original draft. HMH supervised the project, led data discussions,

and revised the manuscript. All authors approved the final version of the manuscript.

**Competing interests.** The authors declare that they have no conflict of interest.

**Disclaimer.** Copernicus Publications remains neutral with regard to jurisdictional claims in published maps and
institutional affiliations.

**Acknowledgments.** The authors acknowledge Hsu-Hung Lee and Ting-Yu Chen for their assistance with field observations, filter sampling, and preliminary data analysis. We also thank Prof. Jr-Chuan Huang from the Department of Geography, National Taiwan University, for IC instrumentation support, and the administration of the
Xitou Experimental Forest, College of Bio-Resources and Agriculture, National Taiwan University, for local site support. AI tools (ChatGPT and Claude) were used to improve the clarity of language; all scientific content was developed and verified by the authors.

**Financial support.** This study was supported by the National Science and Technology Council of Taiwan (Grant
Nos. 112-2111-M-002-014 and 113-2111-M-002-012). W.-C. Lee received support from the National Science and Technology Council of Taiwan (Grant No. 113-2811-M-002-114).



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



**Supporting Information**

S1. Particle concentration estimated by FTIR analysis

S2. Calculation of mass-weighted particle diameter ($D_m$)

S3. Calculation of concentration-weighted $\delta^{15}$N-NH$_4^+$, $\delta^{15}$N-NO$_3^-$, and $\delta^{18}$O-NO$_3^-$

Table S1. Ranges of $\delta^{15}$N-NH$_3$ values (mean ± SD‰) from fertilizer, waste, NH$_3$ slip, and fossil fuel.

Figure S1. (a) Horizontal distance between the sampling location (Xitou) and Taichung, plotted using GeoMapApp (ver 3.7.4). (b) Elevation profile along the same transect between the sampling location (Xitou) and Taichung.

Figure S2. $p$NH$_4^+$ concentration measured by fluorometric versus those estimated by CMAQ analysis.

Figure S3. Estimated $\delta^{18}$O during HNO$_3$ formation adapted from Chen et al. Yellow boxes indicate OH directly derived from the oxygen atom in H$_2$O.

Figure S4. Size-resolved aerosol (a) NH$_4^+$, (b) NO$_3^-$, (c) SO$_4^{2-}$, and (d) BC concentrations.

Figure S5. Particulate $\delta^{15}$N-NH$_4^+$ measured in this study (red symbol) and reported in the literature.

Figure S6 Scatter plot of [NO$_3^-$] + 2*[SO$_4^{2-}$] versus [NH$_4^+$] measured during the observation period.

Figure S7. Resulting concentration of (a) NH$_3$, (b) $p$NH$_4^+$, and (c) the estimated $f$ from fluorometric and CMAQ analysis.

Figure S8. Source apportionment results from the MixSIAR framework for (a) fossil fuel, (b) NH$_3$ slip, (c) waste, and (d) fertilizer.

Figure S9. Particulate $\delta^{15}$N-NO$_3^-$ measured in this study (red symbol) and reported in the literature.

Figure S10. Particulate $\delta^{18}$O-NO$_3^-$ measured in this study (red symbol) and reported in the literature.

Figure S11. Daily size-resolved isotopic composition of $p$NO$_3^-$ from 17 to 20 April.

Figure S12. Daily size-resolved isotopic composition of $p$NO$_3^-$ from 21 to 24 April.

Figure S13. Particulate $\delta^{15}$N-NO$_3^-$ versus $\delta^{18}$O-NO$_3^-$ measured in this study (red symbols) and reported in the literature.

Figure S14. Contributions of nitrate formation pathways estimated by MixSIAR. (a, c) Measured $\delta^{18}$O-NO$_3^-$ values with the corresponding $\delta^{18}$O ranges of potential formation pathways. (b, d) estimated fractional contributions of these pathways (*c.f.*, Fig. S3).

Figure S15. Correlation between daily-averaged CO concentration and $\delta^{18}$O-NO$_3^-$.