# Peer review of "Size-resolved isotope analysis reveals anthropogenic reactive nitrogen transport and transformation in Taiwan mountain forests"

_EGUsphere, 2025_

## Referee Comment (RC1)

Reviewer ID#
Manuscript No.: egusphere-2025-2950
Author: Wen-Chien Lee et al.,
The title of manuscript: **Size-resolved isotope analysis reveals anthropogenic reactive nitrogen transport and transformation in Taiwan mountain forests**

**General comments:**

The authors investigate the transport and transformation of atmospheric reactive nitrogen (Nr), specifically particulate ammonium ($pNH_4^+$) and nitrate ($pNO_3^-$), in a mountain forest environment at Xitou, Taiwan, using size-resolved isotope measurements combined with Bayesian modeling. Samples were collected over a one-week period from April 7 to April 14, 2021, with daytime (09:00–17:00 LT) and nighttime (18:00–06:00 LT) samples analyzed separately. During this period, approximately 26 hours of fog occurred under stagnant atmospheric conditions.

Size-segregated isotope analyses were conducted on these diurnally collected filter samples, although such measurements are technically challenging because of low particle concentrations. The $\delta^{15}N$ and $\delta^{18}O$ isotope measurements were performed using gas chromatography–isotope ratio mass spectrometry (GC-IRMS), and the resulting isotope data were used to infer source contributions and nitrate formation pathways through Bayesian analysis.

Based on the observation that both $\delta^{15}N–NH_4^+$ and $\delta^{15}N–NO_3^-$ decreased during the extended fog event, the authors suggest that the decreasing was associated with fog processes. They further suggest that urban plumes retain $O_3$-driven oxidation signatures, whereas nitrate formed locally in the mountain environment shows greater influence from $RO_2$-involved chemistry and biogenic contributions. Bayesian modeling further indicates that 50–83% of $NH_3$ emissions are combustion-related, while 42–95% of $pNO_3^-$ is attributed to $RO_2$-initiated oxidation during daytime and 6–84% to heterogeneous reactions at night.

However, in comparison with previous studies conducted at the same site and in the same region and other locations worldwide (**Fig S9, S10**), Lee et al. (2025) largely corroborates earlier conclusions regarding urban influence, fog/cloud processing, and secondary oxidation involving $RO_2/O_3$ and isotopic dilution during transport but provides limited additional mechanistic insight. The isotopic effects induced by fog/cloud aqueous-phase processing and those associated with long-range transport act in the same direction, making it difficult to disentangle their respective contributions based on the presented data.

Furthermore, the Bayesian source apportionment using MixSIAR relies on multiple layers of assumptions, including (1) the recalculation of $\delta^{15}N–NH_3$ from particulate $NH_4^+$ based on assumed fractionation factors ($\varepsilon(NH_3–NH_4^+)$), temperature dependence, gas–particle equilibrium, and the approximation of an open system as equilibrium, and (2) the selection of source endmember isotope signatures. Consequently, the posterior distributions are broad and show substantial overlap, between clear & foggy conditions, during both daytime & nighttime (**Fig. 5 (b)**). Although the modeling framework is internally consistent, it does not yield unique or well-constrained source attribution or oxidation pathway estimates.

In addition, the linkage between fog events and isotopic variations appears weak, based on **Fig. 2**. Although uncertainties are not explicitly provided, low values of $\delta^{15}N–NH_4^+$, $\delta^{15}N–NO_3^-$, and $\delta^{18}O–NO_3^-$ also occur outside foggy periods. Therefore, the observed decreases in isotopic values are not unique to fog events, contrary to the authors' interpretation.

Finally, there is room to improve the overall organization and logical flow of the manuscript. The presentation is overly dense in places and lacks clear structural connections between sections, which hinders the clarity of the scientific narrative.

Overall, the study primarily reinforces existing understanding rather than advancing new conceptual or methodological developments in atmospheric nitrogen isotope analysis. Therefore, it may not be well suited for publication in *ACP*, and submission to a more appropriate journal is recommended.

**Specific comments**
- L134-L136: Please clarify the primary scale used for $\delta^{15}N$ and $\delta^{18}O$ (e.g., air-$N_2$, VMOW or others). It is recommended to include the weblink for both international isotope reference materials (USGS 34 and IAEA-NO3)

- L181-L199: The description in this section is confusing and difficult to follow.

- L 220- L227: Please provide the uncertainty for individual measurements and individual data points shown in Fig.2.

- L260-263: The content is somewhat confusing and could benefit from clarification.

- L 340-344: Despite the use of size-resolved isotope measurements, no clear or systematic dependence of $\delta^{15}N$ in $pNH_4^+$ or $pNO_3^-$ on particle size is observed (Fig. 6a,b), which limits the interpretive value of the size-segregated data.